# JSTMapp: A web-based joint spatiotemporal modelling and mapping application for epidemiologists

Alfred Ngwira[1,2,3]*, Samuel O.M. Manda[4], Esron D. Karimuribo[1,3], Sharadhuli I. Kimera[1,3]

**1** Department of Veterinary Medicine and Public Health, Sokoine University of Agriculture, Morogoro, Tanzania, **2** Department of Basic Sciences, Lilongwe University of Agriculture and Natural Resources, Lilongwe, Malawi, **3** SACIDS Foundation for One Health, SACIDS Africa Centre of Excellence for Infectious Diseases, Sokoine University of Agriculture, Morogoro, Tanzania, **4** Department of Statistics, University of Pretoria, Pretoria, South Africa

* alfred.ngwira@sacids.org

## Abstract

Disease mapping models help create disease risk maps, which public health policymakers can use to design disease control and monitoring programmes. These models are now routinely implemented using spatial statistical software packages that use frequentist estimation methods, such as SaTScan and HDSpatialScan, and Bayesian estimation methods, such as the Windows version of Bayesian inference using Gibbs sampling (WinBUGS) and R integrated nested Laplace approximation (INLA). We aimed to develop a user-friendly joint disease spatiotemporal modelling and mapping application (JSTMapp) for epidemiologists and health statistics analysts based on Bayesian methods. Using the R package Shiny and utilising the proven and embedded joint spatial modelling technology in the Bayesian statistical software INLA, we developed the JSTMapp. To illustrate its usage, we used cattle bovine tuberculosis (BTB) and human extrapulmonary tuberculosis (EPTB) data in Africa. The application enables the estimation, mapping, and visualisation of both disease-specific and general spatial and temporal risk factors. It also can evaluate spatial, temporal and spatiotemporal correlations. Additionally, exploratory analyses can be performed, such as mapping the standardised disease incidence ratio. The application showed improved performance when launched from GitHub R as opposed to online from the Shiny server. Improving performance from online servers may seek to use personal servers other than Shiny.

## Introduction

Joint spatiotemporal modelling involves analysing multiple outcome data collected at the exact geographical location and time. The purpose is to establish the spatiotemporal distribution and correlation between diseases. This information is vital for investigating

**Data availability statement:** The data are available from figshare at https://doi.org/10.6084/m9.figshare.30172651.v1.

**Funding:** This study was supported by the Regional Scholarship and Innovation Fund (RSIF) of the Partnership for Skills in Applied Sciences, Engineering and Technology (PASET) (Project Grant No. P165581) grant to SACIDS Africa Centre of Excellence for Infectious Diseases of Humans and Animals in Southern and East Africa (SACIDS-ACE) at the Sokoine University of Agriculture (SUA). Alfred Ngwira was a recipient of an RSIF-PASET doctoral scholarship at SUA.

**Competing interests:** The authors have declared that no competing interests exist.

aetiological factors and designing integrated disease control and surveillance programmes. In this context, the risk factors of one disease are inferred from the risk factors of another disease that exhibits a similar pattern [1]. Most of the data is typically areal, aggregated at a geographic unit such as a county, although point-level data is also employed [2]. Common joint spatial modelling approaches include the multivariate conditional autoregressive (MCAR) model and the shared component construction [3,4]. The MCAR model aims to estimate and test for pairwise disease correlation, whereas the latter assumes correlation exists but does not test for it [4]. The commonly used software for joint disease modelling and mapping includes the Windows version of Bayesian inference using Gibbs sampling (WinBUGS) and R integrated nested Laplace approximation (INLA) [2].

Challenges faced by epidemiologists in joint disease modelling and mapping include knowledge of the theory behind models [5], as they are not part of the conventional statistics. Another challenge is the availability of data, particularly in animal health, which is hindered by poor disease surveillance systems, lack of political will, and insufficient funding for research [6]. Inadequate data analytical skills, especially in programming languages, are further issues faced by many epidemiologists [7,8]. Consequently, using frequently used software for joint disease modelling and mapping [2] requires programming knowledge that most epidemiologists lack.

One way to assist ordinary epidemiologists with limited analytical skills is the development of a user-friendly analytical interface that does not require much programming knowledge. Additionally, the interface may also simplify teaching for students with no or minimal programming background [9]. This may be facilitated by the interest students develop when statistics are embedded into technology [9]. While there has been progress in the development of user-friendly applications for spatiotemporal disease modelling and mapping [10,11], limitations remain in the area of mapping multiple diseases. We set out to develop a joint spatiotemporal modelling and mapping application (JSTMapp) based on the R-INLA code. Cattle bovine tuberculosis (BTB) and human extrapulmonary tuberculosis (EPTB) data in Africa were used for illustration. The JSTMapp may also be applicable for non-health-related outcomes.

## Materials and methods

### Modelling

The number of disease cases $y_{it}^d$ of disease $d = 1, 2, 3, \ldots, D$ from region $i$ and time $t$ are assumed to be distributed as Poisson with mean $\mu_{it}^d = e_{it}^d \theta_{it}^d$, where $e_{it}^d$ is the expected number of cases and $\theta_{it}^d = \mu_{it}^d / e_{it}^d$ is the relative risk. The $e_{it}$ are computed by multiplying the population at risk in each region by the disease incidence rate in the standard population [12], where the standard population is the population for the entire study area, including all regions. If at time $t$, $n_{it}$ is the population at risk in the region $i$, and $r_t$ is the disease incidence rate in the standard population, the expected value in the region $i$ is defined as $e_{it} = n_{it} r_t$. The disease incidence rate in the standard population at a given time is estimated by the ratio of the total observed cases to the total population at risk. The joint spatiotemporal model of multiple diseases [13,14] is defined as:

$$\log\left(\theta_{it}^d\right) = \alpha^d + s_i^d + \omega_s^d s_i + \tau_t^d + \omega_\tau^d \tau_t + \rho_{it}^d + \omega_\rho^d \rho_{it}, \tag{1}$$

where $\alpha^d$ are the disease-specific intercepts, and $s_i^d$, $\tau_t^d$ and $\rho_{it}^d$ are disease-specific spatial, temporal and spatiotemporal effects, respectively. The $s_i$, $\tau_t$ and $\rho_{it}$ are the shared spatial, temporal and spatiotemporal effects, respectively. Parameters $\omega_s^d$, $\omega_\tau^d$ and $\omega_\rho^d$ are the weights of the shared effects. Currently, the JSTMapp is designed to permit the user enter cases of two diseases $d = 1, 2$.

The prior distribution of the shared and disease-specific spatial effects is the intrinsic conditional autoregressive (ICAR) [15]:

$$s_i | s_j, \sigma_s^2, \varnothing \sim Normal \left( \frac{\sum_j \varnothing_{ij} s_j}{\sum_j \varnothing_{ij}}, \frac{\sigma_s^2}{\sum_j \varnothing_{ij}} \right). \tag{2}$$

The $s_j$ are the spatial effects other than $s_i$, while $\sigma_s^2$ and $\varnothing$ are the spatial variance and spatial neighbourhood matrix, respectively. The elements of the neighbourhood matrix are the binary indicators:

$$\varnothing_{ij} = \begin{cases} 1 \text{ if } s_i \text{ and } s_j \text{ are adjacent neighbours,} \\ 0 \quad \text{otherwise.} \end{cases} \tag{3}$$

The shared and disease-specific temporal effects are assigned the first-order random walk (RW1) with a weight matrix $\varphi$, which defines the temporal neighbourhood structure. For a set of temporal effects $\tau_t$ at equally spaced time points $t$, the first-order random walk [16] is specified as:

$$\tau_t | \tau_k \sim \begin{cases} Normal\left(\tau_{t+1}, \sigma_\tau^2\right) \text{ for } t = 1, \\ Normal\left(\frac{\tau_{t-1}}{2} + \frac{\tau_{t+1}}{2}, \frac{\sigma_\tau^2}{2}\right) \text{ for } t = 2, 3, 4, \ldots, T-1, \\ Normal\left(\tau_{t-1}, \sigma_\tau^2\right) \text{ for } t = T. \end{cases} \tag{4}$$

The $\tau_k$ are the temporal effects apart from $\tau_t$, and $\sigma_\tau^2$ is the temporal variance. The expression for RW1 (4) can also be written as an ICAR [14,16]:

$$\tau_t | \tau_k \sim Normal \left( \frac{\sum_k \varphi_{tk} \tau_k}{\sum_k \varphi_{tk}}, \frac{\vartheta_\tau}{\sum_k \varphi_{tk}} \right) \text{ for } t = 1, 2, 3, \ldots, T, \text{ where}$$

$$\varphi_{tk} = \begin{cases} 1 \text{ if } k = t-1 \text{ or } k = t+1, \\ 0 \text{ otherwise,} \end{cases} \tag{5}$$

are temporal weights forming the adjacency matrix $\varphi$, and $\vartheta_\tau$ is the precision of $\tau_t$. The spatiotemporal term is assigned the normal distribution $\rho_{it} \sim N(0, \sigma_\rho^2)$, where $\sigma_\rho^2 = 1/\vartheta_\rho$ is the variance and $\vartheta_\rho$ is the precision. The log-Normal distribution with zero mean and 1/5.9 precision is used to model weights of the shared effects [14]. The precision or inverted variance parameters are assigned either a logGamma (0.5,0.0005), uniform (0,10) or half-Cauchy with a scale parameter equal to 25 [14,17]. While the uniform is considered to be less informative than the log-Gamma, the half-Cauchy is robust to changes in the hyperparameters compared to the log-Gamma [14,17,18]. The fixed effects such as the intercepts are assigned a normal distribution with zero mean and 0.001 precision.

Model estimation involves estimating a vector of the main parameters $\Omega$, such as spatial, temporal and spatiotemporal effects, including weights of the shared effects [19]. It also involves estimating the vector of variance or precision parameters $\Psi$ for all outcomes. This is done by computing the joint posterior distribution of the two sets of parameters given the stacked data $y$ for all outcomes defined as:

$$p\left(\Omega, \Psi \middle| y\right) \propto p\left(y \middle| \Omega, \Psi\right) p\left(\Omega \middle| \Psi\right) p(\Psi), \tag{6}$$

where $p\left(y \middle| \Omega, \Psi\right)$ is the conditional data likelihood, $p\left(\Omega \middle| \Psi\right)$ is the conditional prior of $\Omega$ given $\Psi$ and $p(\Psi)$ is the prior of $\Psi$. Estimation of the joint posterior is through the integrated nested Laplace approximation (INLA) [20], where the marginal distributions of specific parameters $p\left(\Omega_u \middle| y\right)$ for $u = 1, 2, 3, \ldots, U$ and $p(\Psi_s|y)$ for $s = 1, 2, 3, \ldots, S$ are computed. The expressions for the posterior marginals are specified as:

$$p\left(\Omega_u \middle| y\right) = \int p(\Omega_u | \Psi, y) p\left(\Psi \middle| y\right) d\Psi,$$

$$p\left(\Psi_s \middle| y\right) = \int p(\Psi | y) d\Psi_{-s}. \tag{7}$$

Once the posterior marginals are obtained, posterior summaries such as the mean and standard deviation are computed.

To have smooth model-based risk maps, the inverse distance weighting (IDW) interpolation is implemented [21], where the estimated risk values $\theta_{it}$ from the fitted model are used as input observed values during spatial interpolation. Assuming for each time point $t$, $(x_i, y_i)$ for $i = 1, 2, 3, \ldots, n$ are the centroids of grid locations that have model-based risk $\theta_{it}$, the estimate of the risk at unobserved grid location $(x_j, x_j)$ for $j = 1, 2, 3, \ldots, J$ by IDW is defined as:

$$\hat{\theta}_t\left(x_j, x_j\right) = \sum_{i=1}^{n} \omega_t\left(\left(x_i, x_i\right), \left(x_j, x_j\right)\right) \theta_t(x_i, x_i), \tag{8}$$

where $\omega_t\left(\left(x_i, x_i\right), \left(x_j, x_j\right)\right)$ is the weight of each data point in estimating the unknown value and is specified as:

$$\omega_t\left(\left(x_i, x_i\right), \left(x_j, x_j\right)\right) = \frac{d_t^{-p}\left(\left(x_i, x_i\right), \left(x_j, x_j\right)\right)}{\sum_{i=1}^{n} d_t^{-p}\left(\left(x_i, x_i\right), \left(x_j, x_j\right)\right)}. \tag{9}$$

The quantity $d_t\left(\left(x_i, x_i\right), \left(x_j, x_j\right)\right)$ is the Euclidean distance between the known $(x_i, x_i)$ and unknown grid point $(x_j, x_j)$, where $p$ is the power for control and is set to 2, which is considered to be the frequently used value [21]. The dimension of the prediction grid is set at $10000 \times 10000$, since this tends to produce a smoother risk surface [21].

## Components of JSTMapp

**Input bar.** This bar has input tabs where the disease case data in csv format is uploaded. Data variables such as area and time are also selected from this sidebar. The shapefile data file with extension shp and all associated files are also uploaded in this input bar. The input sidebar has additional input tabs, which are conditional to the model estimation tab, where the user can select the number of covariates, and model type, including specifying the prior for precision parameters.

**Output bar.** The output bar has six tabs, which include "Explore", "Model estimation", "Spatial and temporal risk", "Spatiotemporal risk", "Prediction" and "Correlation". In the "Explore" tab, a map of the standardised raw incidence ratios and a time series of the observed cases are plotted. Model estimation is performed under the "Model estimation" tab, while spatial, temporal and spatiotemporal risk is displayed under the "Spatial and temporal risk" and "Spatiotemporal risk" tabs, respectively. The "Prediction" tab is designed to show the predicted raw incidence ratio and the model-based risk. Evaluation of spatial, temporal and spatiotemporal correlation of weights of the shared effects using Pearson correlation is conducted under the "Correlation" tab. Displaying of outputs from the tabs after the "Model estimation" tab is dependent on the outputs from the "Model estimation" tab, otherwise, they do not show.

**Installation.** The JSTMapp can be run in R/RStudio software by launching it from the GitHub repository at https://github.com/alfredngwira/JSTMapp. This can be done by running the following command:

```
>shiny::runGitHub("alfredngwira/JSTMapp",subdir="inst/JSTMapp")
```

Alternatively, the user can access the JSTMapp's interface in R/RStudio by executing the following commands:

```
>install.packages("devtools")
>devtools::install_github("alfredngwira/JSTMapp", ref="main")
>library(JSTMapp)
>run_app()
```

To use the JSTMapp in R/RStudio, the R packages shiny, INLA, ggplot2, gridExtra, RColorBrewer, Hmisc, shinyjs, dplyr, spdep, raster, tmap, gstat, dotwhisker and sn must be installed. A description of these R packages in terms of how they function when using JSTMapp is shown in Table 1.

The JSTMapp user interface can also be accessed online from the Shiny server at https://alfredngwira.shinyapps.io/JSTMapp. The online version uses the server computer and it tends to be associated with memory problems, especially when the free payment plan is utilised. We recommend using the application in R/RStudio by launching it from GitHub, where the local computer performs the computation, thereby minimising system memory issues.

## Illustrative example

### Sources of data

In this section, we demonstrated the usage of JSTMapp by analysing cattle bovine tuberculosis (BTB) and human extrapulmonary tuberculosis (EPTB) case data for African countries from 2005 to 2018. The choice of cattle BTB and human EPTB was due to their epidemiological relationship [35] and the scarcity of laboratory-diagnosed human zoonotic TB cases. We considered the study period 2005–2018 since case data for cattle BTB were available only during this time frame. The cattle BTB data were accessed from the World Organisation for Animal Health Information System (WAHIS) [36], while human TB data were obtained from the World Health Organisation (WHO) [37]. From the human TB data file, in addition to EPTB, we also collated the total number of human TB cases per country and per year, which served as the total population at risk. Data were also compiled on total cattle population, including number of slaughtered cattle for each country and year from the Food and Agriculture Organisation (FAO) [38]. The data on the number of slaughtered

**Table 1. Description of R packages that are used by JSTMapp.**

| Name of package | Use in JSTMapp | References |
|---|---|---|
| shiny | Develop and run the web-based application | [22] |
| INLA | Fit joint spatiotemporal model | [20] |
| ggplot2 | Plot temporal graphs | [23] |
| gridExtra | Arrange multiple graphs in one panel | [24] |
| RColorBrewer | Define colours for risk maps | [25] |
| Hmisc | Generate correlation matrix with *p*-values | [26] |
| shinyjs | Enable JavaScript operations | [27] |
| dplyr | Summarise data | [28] |
| spdep | Create a neighbourhood map | [29] |
| raster | Upload shapefiles | [30] |
| tmap | Plot maps | [31] |
| gstat | Perform IDW interpolation | [32] |
| dotwhisker | Plot dot whisker | [33] |
| sn | Required by INLA for posterior sampling | [34] |

cattle was used as cattle population at risk of BTB, considering that most BTB cases are diagnosed at abattoirs through postmortem examination. We also collected human population data for each country and year from the World Bank [39], including the average temperature and precipitation data for each country from WorldClim [40]. Using land area in km² [41], cattle and human population for each country, we calculated densities by dividing population by land area. Population density, average temperature and precipitation were used as covariates. Our approach in modelling national and annual level outcome data on climatic and weather variables was based on the literature [42]. The shapefile data for Africa was downloaded from the ArcGIS hub website [43]. The objective in this case study was to examine the joint spatiotemporal trends of cattle BTB and human EPTB so as to determine potential countries where zoonosis was likely. The results would be useful to the African Centers for Disease Control and Prevention (CDCP) in aiding policy guidance regarding countries that need to strengthen integrated One Health strategies to reduce the TB burden in both humans and animals.

**Exploratory analysis**

Fig 1 displays the "Explore" tab presenting a map of the standardised raw incidence ratio, alongside a time series of the number of cases for cattle BTB and human EPTB in Africa from 2005 to 2018. A few isolated countries in the central, southeastern and northern parts of Africa exhibited a high incidence ratio concerning cattle BTB. A few countries with high incidence ratio concerning human EPTB were in the central eastern and central northern Africa. High incidence ratio patterns common to both cattle and humans were observed in Algeria and Tunisia. The temporal distribution of cattle BTB and human EPTB cases demonstrated a similar trend, peaking in 2011.

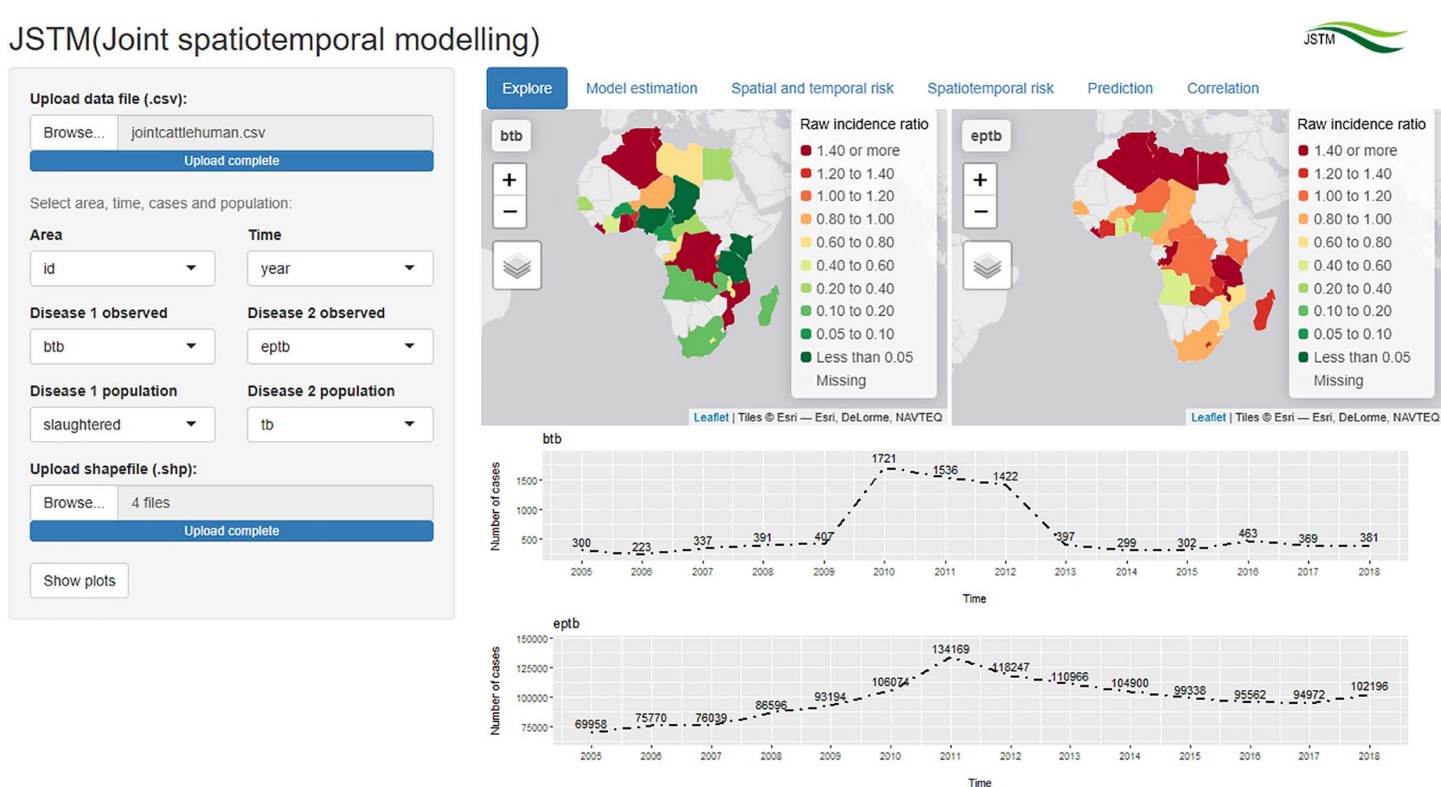

**Fig 1. "Explore" tab showing map of raw incidence ratio and time series of cattle BTB and human EPTB in Africa (Source of shapefiles: Arc-GIS [43]).**

## Model estimation

The raw summaries of the estimated model, including the dot and whiskers plot of the estimated parameters are displayed in the "Model estimation" tab (Fig 2). The intercepts on the exponential scale indicated a significant reduced overall risk of cattle BTB (0.371; 95% credible interval (CI): 0.184, 0.742) and a significantly greater than average risk of human EPTB (1.209; 95% CI: 1.027, 1.423). Based on the inverse of the estimated precision parameters (1/precision), the disease-specific spatial random variation of cattle BTB was greater than that of human EPTB. The weights of the shared spatial risk indicated a greater contribution of cattle BTB random factors to the shared spatial risk compared to human EPTB random factors. Similarly, there was a greater contribution of cattle BTB temporal random factors than human EPTB temporal random factors to the shared temporal risk. Incorporation of covariates is demonstrated in a supplementary figure S1 Fig. Cattle density was negatively associated with bovine tuberculosis, while human density was positively associated with bovine tuberculosis. Average annual temperature was negatively associated with human extrapulmonary tuberculosis.

## Spatial and temporal risk

Results from the "Spatial and temporal risk" tab are shown in Fig 3. In this tab, the spatial risk pattern for cattle BTB indicates that few countries in the central, southeast and northwest of Africa were at high risk. The spatial risk pattern for human EPTB reveals that countries in the central east, north and west of Africa were at high risk. The shared risk patterns between cattle BTB and human EPTB were descending from north to south. There was a decreased pattern in the temporal risk of cattle BTB from 2005 to 2013, followed by an increased pattern thereafter. The risk for human EPTB remained consistently on average throughout the study period.

## Spatiotemporal risk

Model-based spatiotemporal risk patterns of cattle BTB and human EPTB are shown in the "Spatiotemporal risk" tab (Fig 4). The patterns for BTB in cattle indicated that countries such as Algeria, Tunisia, Democratic Republic of Congo (DRC) and Mozambique were consistently at higher risk over time, and similar patterns were observed for human EPTB.

## Prediction

The "Prediction" tab (Fig 5) shows the spatially predicted patterns in terms of both the raw incidence ratio and the model-based estimated risk. The predicted risk of human EPTB was higher in the east and north of Africa, while the predicted risk pattern for cattle BTB was generally descending from north to south of Africa.

## Correlation

Spatial, temporal and spatiotemporal correlation between cattle BTB and human EPTB are displayed in the "Correlation" tab (Fig 6). There was significant positive spatial and temporal correlation between cattle BTB and human EPTB.

## Discussion

This study has developed and implemented a user-friendly interface JSTMapp for joint spatiotemporal modelling and mapping of two diseases. The web-based spatial application JSTMapp uses the R packages Shiny and INLA to implement a bivariate spatial random-effects model. This model is specifically designed to analyse spatial lattice count data, which are modelled as Poisson random variables. The case study involved a retrospective analysis of cattle bovine tuberculosis and human extrapulmonary tuberculosis cases for Africa from 2005 to 2018. The JSTMapp facilitates exploratory data analysis and fitting of the multivariate shared component spatiotemporal model, including displaying the estimated risk as spatially interpolated patterns. The application has shown better performance when run in R/RStudio from GitHub as opposed to online from the Shiny server.

# JSTM(Joint spatiotemporal modelling)

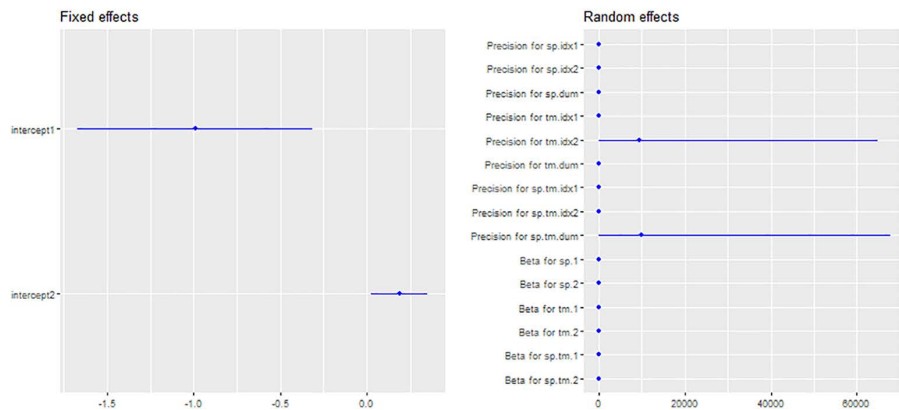

**Upload data file (.csv):**

| Browse... | jointcattlehuman.csv |

Upload complete

Select area, time, cases and population:

**Area**
id ▼

**Time**
year ▼

**Disease 1 observed**
btb ▼

**Disease 2 observed**
eptb ▼

**Disease 1 population**
slaughtered ▼

**Disease 2 population**
tb ▼

Select number of covariates, model and precision prior:

**Covariates**
- ● None
- ○ One
- ○ Two
- ○ Three
- ○ Four

**Model**
- ○ Spatial
- ○ Temporal
- ○ Spatial + Temporal
- ● Spatiotemporal

**Precision prior**
- ○ LogGamma
- ○ Uniform
- ● Half-Cauchy

Select covariates (optional):

**Covariate 1**
▼

**Covariate 2**
▼

**Covariate 3**
▼

**Covariate 4**
▼

**Upload shapefile (.shp):**

| Browse... | 4 files |

Upload complete

Summary of results

---

Explore | **Model estimation** | Spatial and temporal risk | Spatiotemporal risk | Prediction | Correlation

```
Time used:
    Pre = 1.19, Running = 36.5, Post = 0.71, Total = 38.4
Fixed effects:
            mean    sd 0.025quant 0.5quant 0.975quant  mode kld
intercept1 -0.99 0.346     -1.672    -0.99     -0.313 -0.99   0
intercept2  0.19 0.082      0.028     0.19      0.352  0.19   0

Random effects:
  Name      Model
    sp.idx1 Besags ICAR model
    sp.idx2 Besags ICAR model
    sp.dum Besags ICAR model
    tm.idx1 Besags ICAR model
    tm.idx2 Besags ICAR model
    tm.dum Besags ICAR model
    sp.tm.idx1 IID model
    sp.tm.idx2 IID model
    sp.tm.dum IID model
    sp.1 Copy
    sp.2 Copy
    tm.1 Copy
    tm.2 Copy
    sp.tm.1 Copy
    sp.tm.2 Copy

Model hyperparameters:
                            mean       sd 0.025quant 0.5quant 0.975quant      mode
Precision for sp.idx1      0.085 4.10e-02      0.030    0.077   1.87e-01     0.063
Precision for sp.idx2      2.115 1.96e+00      0.290    1.552   7.30e+00     0.778
Precision for sp.dum       0.106 5.87e-01      0.001    0.019   7.29e-01     0.001
Precision for tm.idx1      7.521 6.22e+00      1.451    5.790   2.40e+01     3.489
Precision for tm.idx2   9503.220 5.91e+04     88.023 1468.211   6.48e+04   186.923
Precision for tm.dum      53.076 1.45e+03      0.002    1.105   2.81e+02     0.000
Precision for sp.tm.idx1   1.690 2.52e-01      1.245    1.672   2.23e+00     1.639
Precision for sp.tm.idx2  55.010 1.14e+01     34.369   54.349   7.90e+01    53.993
Precision for sp.tm.dum 9942.029 5.70e+04     92.130 1704.883   6.79e+04   191.890
Beta for sp.1              0.375 5.31e-01      0.030    0.218   1.70e+00     0.079
Beta for sp.2              0.153 2.46e-01      0.010    0.082   7.45e-01     0.027
Beta for tm.1              0.364 9.90e-01      0.003    0.108   2.31e+00     0.002
Beta for tm.2              0.088 2.57e-01      0.001    0.027   5.53e-01     0.002
Beta for sp.tm.1          8.848 8.49e+01      0.024    0.779   5.81e+01     0.039
Beta for sp.tm.2          1.132 3.94e+00      0.006    0.273   7.60e+00     0.005

Deviance Information Criterion (DIC) ...............: 3287.22
Deviance Information Criterion (DIC, saturated) ....: 700.92
Effective number of parameters .....................: 336.34

Marginal log-Likelihood:  -2273.23
CPO, PIT is computed
Posterior summaries for the linear predictor and the fitted values are computed
(Posterior marginals needs also 'control.compute=list(return.marginals.predictor=TRUE)')
```

**Fig 2. "Model estimation" tab showing summaries of the estimated model of cattle BTB and human EPTB in Africa.**

---

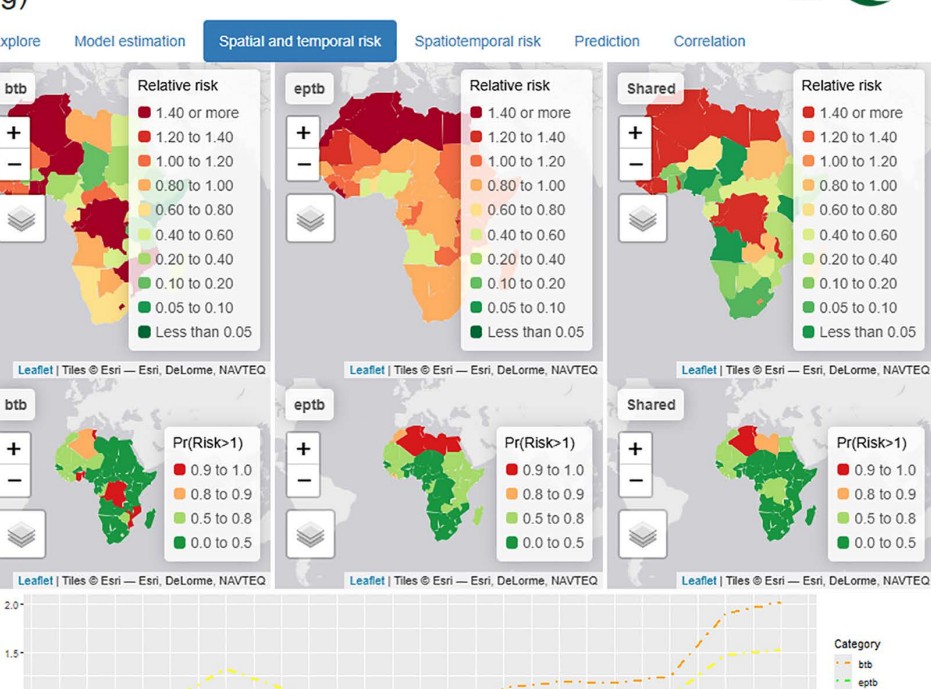

**Fig 3. "Spatial and temporal risk" tab showing spatial and temporal risk of cattle BTB and human EPTB in Africa (Source of shapefiles: Arc-GIS [43]).**

The advantage of using JSTMapp over the traditional R programming approaches is the ease of data analysis, especially for non-experts in programming. This makes it simpler for epidemiologists to analyse data and for university lecturers to teach students [9]. Another benefit of using JSTMapp is the interactive experience, where the user can dynamically change the data input and instantly see the results. This is not the case with the traditional use of programming, where changes in the code have to be made for any new data input to produce the output. Additionally, unlike previous web-based spatial model applications [10,11], that only fit univariate models, our web application can accommodate multivariate spatial random effects models using shared component methods. Furthermore, JSTMapp is capable of implementing spatial prediction to create smooth risk surfaces, a feature that previous applications do not offer.

The findings and resulting conclusions from the case study should be treated with caution due to various factors that may impact the validity and reliability of the results. The higher risk of individual cattle BTB and human EPTB, as well as the shared cattle BTB and human EPTB in north Africa, may be due to the prolonged dry environmental conditions, which tend to be associated with a higher risk of tuberculosis [44]. According to Xiao et al. [44], dry conditions tend to limit the production of the protective mucus layer in the respiratory tract, thereby increasing vulnerability to tuberculosis infection. Furthermore, the dry conditions in north Africa may result in cattle competing for fewer food and water resources, which, in turn, may increase the risk of cattle tuberculosis infection [45]. A higher risk of human EPTB in east and north Africa may be attributed to the risky practice of drinking cattle blood by Maasai people in east Africa with the aim of supplementing milk and to cure diseases [46]. Furthermore, other tribes in east and north Africa tend to drink sour milk for food and

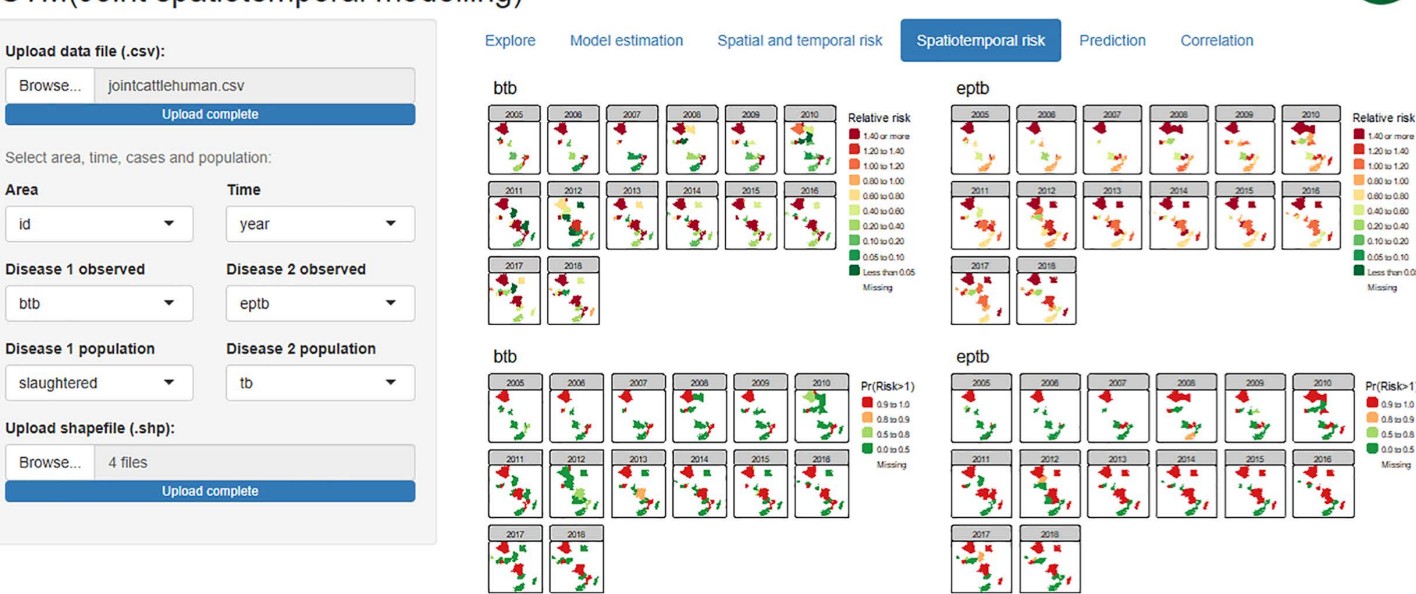

**Fig 4.** "Spatiotemporal risk" tab showing estimated spatiotemporal risk of cattle BTB and human EPTB in Africa (Source of shapefiles: Arc-GIS [43]).

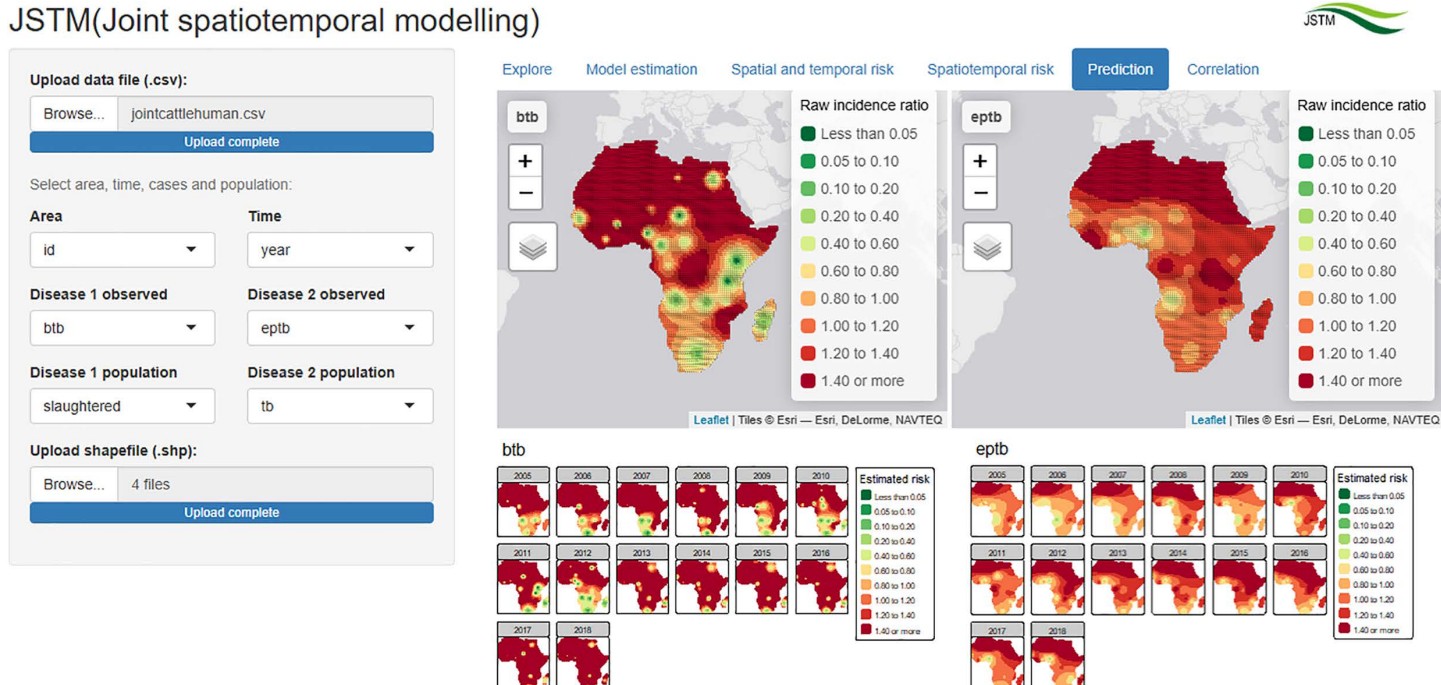

**Fig 5.** "Prediction" tab showing the predicted raw incidence ratio and model-based risk of cattle BTB and human EPTB in Africa (Source of shapefiles: ArcGIS [43]).

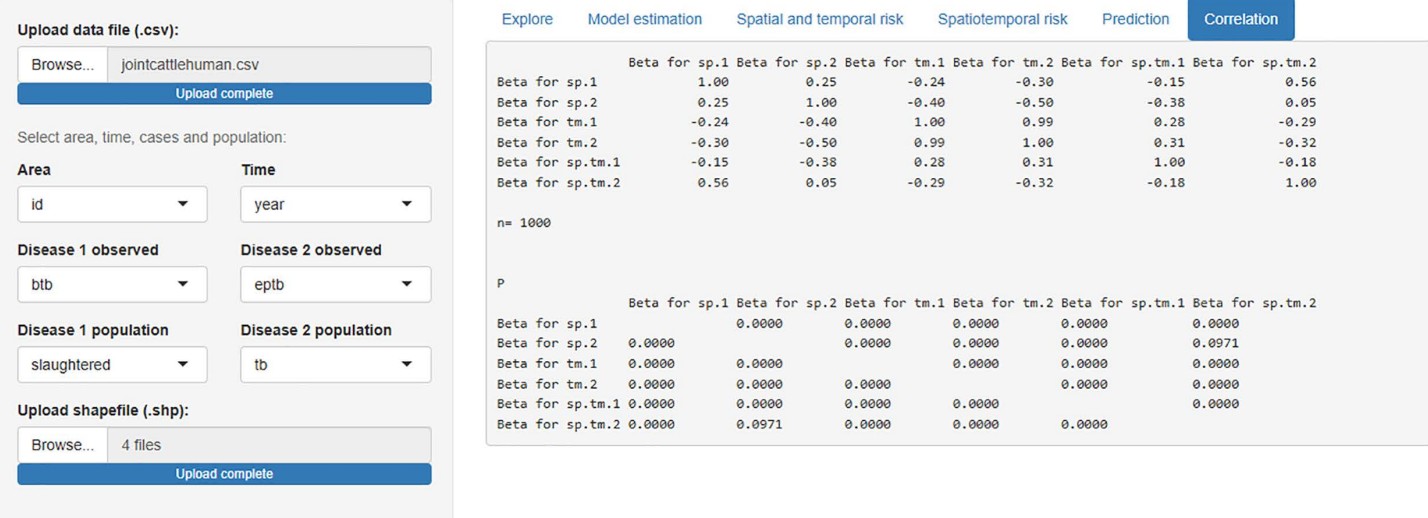

**Fig 6.** "Correlation" tab showing spatial, temporal and spatiotemporal correlation of cattle BTB and human EPTB in Africa.

medicine [47], which poses a risk of zoonotic TB infection that often turns into EPTB. The higher burden trends of human EPTB in east Africa correspond with the literature [48]. The observed high-risk patterns of bovine tuberculosis in low-cattle density areas, such as the central African countries, could imply the influence of other factors, including warm and moist environments which favor *M. bovis* [49]. The positive effect of human density on cattle BTB may be attributed to increased animal movements and trade activities by many people, which, in turn, due to close contacts, risk BTB infection in cattle. The finding of human density as a positive risk factor of cattle BTB in Africa is in agreement with the literature [50], where human density was indirectly and positively correlated with bovine tuberculosis. The negative correlation between temperature and human EPTB may be explained by the longer exposure time when temperatures are low, since individuals tend to stay indoors [51].

Despite the strengths of the application, it lacks flexibility in certain model parameterisations and graphic outputs. Additionally, the application can handle a limited number of predictors. The functionality of forecasting was not incorporated due to implementation challenges. The fact that the application does not function properly from the Shiny server, but performs well in R from GitHub, implies that minimal programming skills are required for installation. This appears to be a common issue with applications hosted online by the Shiny server, as a similar application by Moraga [11] also operates from R. Additionally, for a better spatial and temporal resolution, example data in the case study did not involve a smaller spatial and temporal scale, such as district and month or week, respectively. We aim to improve our efforts by adding more features that will help analyse different outcomes and geostatistical spatial models. Additionally, we plan to expand the variety of spatial distribution options available, moving beyond just the Gaussian conditional autoregressive spatial models.

## Conclusion

Our web-based application, the JSTMapp, improves accessibility to spatial data analysis for epidemiologists by reducing the need for coding skills. With its user-friendly interface, the platform allows users to easily upload their datasets and perform advanced spatial analyses. The visualisation of results is designed to be clear, enabling professionals of all technical levels to quickly identify and interpret areas of high disease risk. This feature greatly supports the generation of public health insights that can inform policy interventions and resource planning.

## Supporting information

**S1 Fig. "Model estimation" tab showing summaries of the estimated model of cattle BTB and human EPTB in Africa.**
(TIF)

## Author contributions

**Conceptualization:** Alfred Ngwira.

**Methodology:** Alfred Ngwira.

**Supervision:** Samuel O.M. Manda, Esron D. Karimuribo, Sharadhuli I. Kimera.

**Writing – original draft:** Alfred Ngwira.

**Writing – review & editing:** Alfred Ngwira, Samuel O.M. Manda, Esron D. Karimuribo, Sharadhuli I. Kimera.

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
