## [Decision Letter · Decision Letter 0]

13 May 2025

Dear Dr. Ngwira,

We look forward to receiving your revised manuscript.

Kind regards,

Rebecca Lee Smith, D.V.M., M.S., Ph.D.

Academic Editor

PLOS ONE

Journal Requirements:

3. Thank you for stating the following in the Acknowledgments Section of your manuscript: [This study was supported by the Regional Scholarship and Innovation Fund (RSIF) of the Partnership for Skills in Applied Sciences, Engineering and Technology (PASET) (Project Grant No. P165581) grant to SACIDS Africa Centre of Excellence for Infectious Diseases of Humans and Animals in Southern and East Africa (SACIDS-ACE) at the Sokoine University of Agriculture (SUA). Alfred Ngwira was a recipient of an RSIF-PASET doctoral scholarship at SUA.]

Please remove any funding-related text from the manuscript and let us know how you would like to update your Funding Statement. Currently, your Funding Statement reads as follows: “The authors received no specific funding for this work.”

4. We note that Figures 1,3,4 and 5 in your submission contain [map/satellite] images which may be copyrighted. All PLOS content is published under the Creative Commons Attribution License (CC BY 4.0), which means that the manuscript, images, and Supporting Information files will be freely available online, and any third party is permitted to access, download, copy, distribute, and use these materials in any way, even commercially, with proper attribution. For these reasons, we cannot publish previously copyrighted maps or satellite images created using proprietary data, such as Google software (Google Maps, Street View, and Earth). For more information, see our copyright guidelines: http://journals.plos.org/plosone/s/licenses-and-copyright.

1. You may seek permission from the original copyright holder of Figures 1,3,4 and 5 to publish the content specifically under the CC BY 4.0 license. 

Reviewers' comments:

Reviewer's Responses to Questions

**Comments to the Author**

1. Is the manuscript technically sound, and do the data support the conclusions?

Reviewer #1: Partly

Reviewer #2: Yes

2. Has the statistical analysis been performed appropriately and rigorously?

Reviewer #1: Yes

Reviewer #2: Yes

3. Have the authors made all data underlying the findings in their manuscript fully available?

Reviewer #1: Yes

Reviewer #2: Yes

4. Is the manuscript presented in an intelligible fashion and written in standard English?

Reviewer #1: Yes

Reviewer #2: Yes

Reviewer #1: Review of the Manuscript:

“JSTMapp: A web-based joint spatiotemporal modeling and mapping application for epidemiologists”

The manuscript focuses on JSTMapp, a user-friendly web-based dashboard for joint spatiotemporal modelling and mapping of two diseases using Bayesian hierarchical modelling applying INLA and R Shiny. The application deals with cattle bovine tuberculosis (BTB) and human extrapulmonary tuberculosis (EPTB) across Africa from 2005 to 2018.

The study presents a highly a welcome tool for epidemiologists who may not have advanced programming skills. By integrating Bayesian spatiotemporal modelling into a graphical interface, JSTMapp facilitates a complex modelling process.

Strengths

• JSTMapp presents a user-friendly application for complex joint spatiotemporal modelling which is highly relevant and valuable, especially for public health researchers with limited statistical or programming expertise.

• The approach implements multivariate, disease specific and multivariate shared components, allowing for the exploration of spatial, temporal, and spatiotemporal correlations.

• The approach incorporates prediction capabilities for a wide range of prediction intervals.

• The approach includes an interactive exploratory data analysis component, a mapping of standardized incidence ratios module, and a spatiotemporal risk interpolation component enhancing user experience.

• The application is well documented with an easily to use installation guide via GitHub.

Limitations and Areas for Improvement

Despite its strengths, the manuscript and the application have several important limitations that should be addressed:

1. Lack of Model Flexibility

The current version of JSTMapp assumes that all spatially structured, temporally structured, and unstructured components are significant contributors to explaining the spatiotemporal variability of the diseases.

• Critique: In real-world applications, not all random components may be significant.

• Suggestion: The approach should allow model selection options (particularly , selecting spatial effects only, temporal effects only, or spatiotemporal effects only) based on model comparison criteria like the DIC, WAIC, or CPO. See inter alia Jaya and Folmer 2020, J. of Geographical Systems, pp 105-142; 2021, Geographical Analysis, pp 767-817.

2. Fixed Prior Specifications

The application does not allow users to adjust prior distributions for the random effects or the hyperparameters (see Jaya and Folmer 2020, J. of Geographical Systems, pp 105-142).

• Critique: The choice of priors can heavily influence Bayesian model prediction results, especially in the case of sparse data.

• Suggestion: Providing options for users to modify priors (particularly, for spatial precision, temporal precision, and weight parameters) would significantly increase the flexibility and broader applicability of JSTMapp.

3. Limited Covariate Inclusion

The model is restricted to area and time effects as covariates. No allowance for additional covariates (such as socio-economic factors, climate variables, etc.) is provided.

• Suggestion: Allowing JSTMapp for the inclusion of external covariates would make it more powerful for real-world disease mapping and risk modelling.

• Note: For an overview of the prediction accuracy of multivariate spatiotemporal models with a confounded covariate versus univariate models without covariates, see Jaya and Folmer, 2025, J. of Geographical Systems, 113-146.

4. Lack of Hotspot Detection

The application does not support posterior probability mapping or hotspot analysis (particularly, mapping of areas where the posterior probability of risk exceeding a threshold is above 0.8 or 0.9). See Jaya, Folmer, and Lundberg, 2024, Annals of Regional Science , pp 107-140, for details)

• Suggestion: Adding this feature would be very useful for public health interventions.

5. Future Prediction Needs Accuracy Evaluation

Although the application enables future risk prediction, it does not evaluate the prediction performance (see Jaya and Folmer 2020, J. of Geographical Systems, pp 105-142)

• Critique: Without validating using a training and testing dataset split, the forecasted risks lack an objective measure of accuracy.

• Suggestion: The authors should consider integrating a model validation procedure where part of the data is used for training and another part for testing. Predictive performance metrics such as the MAE, RMSE, or correlation coefficients between observed and predicted values should be provided as statistics to assess forecast accuracy. (See inter alia Jaya and Folmer 2020, J. of Geographical Systems, pp 105-142; Jaya and Folmer, 2025, J. of Geographical Systems, 113-146)

6. Deployment Issues

The application performs better when run locally rather than online via a Shiny server due to memory limitations.

• Suggestion: While the paper recommends using GitHub local installation, applying Dockerization or lightweight server deployment are likely to improve online usability.

7. Graphics and Interface Improvements

• Critique: Some of the graphical outputs could be improved in terms of better readability (e.g., appropriate scaling legends, improving color schemes for color-blind users).

• Suggestion: Future updates should offer user-friendely color palettes and exportable high-resolution images.

• Note: For high-resolution prediction and mapping the author may consult Jaya and Folmer 2022, J. of Geographical Systems, 525-581.

Minor Points

• Typographical and grammatical editing are needed to improve the clarity and professionalism of the manuscript. Some minor errors in English expression should be corrected.

• It would be beneficial if the authors could provide a video tutorial or sample case studies in the GitHub repository for beginners.

Reviewer #2: The authors describe a bioinformatic application called JSTMapp that they have developed in R, and can be used to fit statistical models to epidemiological surveillance datasets consisting of spatiotemporally referenced case-counts for multiple diseases. JSTMapp can be run using open-source tools such as R, GitHub, or an online interface. They provide an example dataset of national, annual level case-counts of two diseases – BTB, and EPTB – sourced from publicly available repositories and georeferenced to areal units of numerous African countries. When loaded into the JSTMapp interface, the application applies a suite of modeling and data visualization processes to the data allowing the user to explore trends and parameters derived from the model, including spatially interpolated predictions based on shapefiles of the areal units. The example dataset also includes the environmental variables of temperature and precipitation (however it is unclear to me if these are included in the model, as I do not see parameter estimates for them under the “Model estimation” tab). As described, the tool has notable applications for epidemiologists and students of biostatistics, although some upfront work is required of the user to set up the time series data in the right format and link it to the polygons in the shapefile.

I am someone who works with spatiotemporally referenced health data frequently, but who has only intermediate R and basic GitHub proficiency. I have not myself ever fitted the kinds of models employed by JSTMapp (INLA, MCAR etc.) though I am aware of them, have colleagues who work with them frequently, and understand them to be appropriate and state of the art methods for this kind of data. I therefore approach this review from the perspective of the “ordinary epidemiologist with limited analytical skills” that the authors describe.

Introduction:

Not being previously familiar with the term “joint spatiotemporal modeling”, it is unclear to me whether in these kinds of model, each disease outcome serves as a predictor for the others. Is the burden of each disease jointly modeled conditional on the presence of the others. Could the authors please clarify? Is it only applicable to infectious diseases? This is not specified.

Methods:

This section starts with a lengthy explication of the modelling using equations and formal notation. I am not qualified to review these equations, however I do believe that in most journal articles, equations are numbered so that they can be easily referenced within the text. I think the “components of the application” should be moved up to the beginning of the methods section, and the model descriptions should be lower down or even moved to a supplemental appendix.

Results:

I’m not sure that a results section is even required for this type of paper, since you are not offering an interpretation of scientific findings, but rather describing a tool and an illustrative example of its application. I will defer to the editor on this point, but perhaps “illustrative example” would be a better name for this section.

The sources of the illustrative data could perhaps be described in a separate section with its own subheading, and I don’t think that full URLs should be given within the text. Rather, online sources should be cited using in-text citations e.g. (WHO 2025) referencing sources in the bibliography which would include their URLs. Perhaps the sources could be better presented in a small table.

I was able to download the data and JSTMapp R package and run the application using RStudio. I was not able to do this within GitHub as I am less familiar with that platform. The database that’s downloadable from GitHub contains what appear to be several meteorological variables (temp, tmax, tmin, prec). These are static across years within each country in the database and are not described or cited in the article. It is unclear if they are being used as predictors in the model, and in any case, they probably should not be, since national aggregate statistics are not a suitable level of resolution for weather parameters.

Figure 1: I selected the variables in the same way that the authors described, but the visualizations did not appear in exactly the same way. Instead of the maps in the upper left-hand corner of figure 1, I got an error message saying “the following option does not exist: check.and.fix..

Furthermore, the label for figure 1 does not sufficiently describe the map panel. Are these predictions from the INLA component of the model? The legend title is not legible in either the article figure, or in the application interface. I realize that the data is meant to be illustrative, but the authors should perhaps acknowledge that in a real epidemiological analysis, it would not be appropriate to do a spatial interpolation for an entire continent that was based on national level aggregates, since that does not offer sufficient spatial resolution. Similarly, annual totals are probably not a sufficient temporal resolution to derive conclusion about infectious disease transmission. The optimal use case for this package, I think, would be at least monthly or weekly resolution epidemiological data aggregated to province, or district level.

Figure 2: Since the other tabs offer visually appealing visualizations of the model results, could the “Model estimation” tab do the same, rather than displaying raw R model output? Perhaps the parameter estimates could be visualized in dot-and-whisker plots.

Figure 3: The maps in this figure did not appear when I ran the application in RStudio. I got the same error message about the option not existing.

Figure 4: The prediction maps appeared when I ran the application, but the estimation maps did not (same error message). The application does not seem to offer any customizability in the visual outputs and as such is not particularly interactive. More important than generating a series of panels for each year, would be for the user to be able to zoom in and out of the map and click on particular locations to see the prediction values displayed in a pop-out. Furthermore, the user should be able to assign names to the diseases so that they appear in the graph legends rather than “Disease 1” and “Disease 2”.

Figure 5: Again, the maps did not appear when I ran the application. I realize that, since the data is intended to be illustrative, commenting on the actual results of the model is beside the point, however, I do find the post-2018 predictions for EPTB implausible. What possible reason could there be for the relative risk to plummet below the null value of 1 like that? It does not seem to be an extrapolation from the observed trends pre-2018. The authors do not comment on this strange finding.

Figure 6: When I select the “Correlation” tab, no results appear and instead an error message says “Package 'sn' is required to proceed but is not installed. Please install.” sn is not listed among the required packages. Also, the database is named differently in the downloadable file (jointafrica.csv) compared to in the screenshot figures (jointcattlehuman.csv).

Discussion:

The authors are right to highlight the lack of flexibility of the tool. The ideal use case for JSTMapp is so specific that it is hard to imagine it arising frequently enough to warrant a dedicated package, and yet the example dataset that it is tailored to (the BTB, EPTB example) is not even a suitable use case for this kind of model, since the data is aggregated to national and annual totals.

General comments:

The dashboard allows the user to select two diseases and their populations at risk from the database using the dropdown menu. Is it possible to jointly model more than two diseases?

The authors refer to “the application” throughout the manuscript. I think they should own the name JSTMapp more and say e.g. “JSTMapp” can be run in R software…” (line 164). The name is not even mentioned in the abstract and doesn’t occur in the main text until line 178. It’s a decent name and I think the authors should not be hesitant to use it. It would be nice if JSTMapp had a simple logo that could appear on the dashboard and thumbnails for the package. The authors could consider developing one on Canva or a similar platform.

**Do you want your identity to be public for this peer review?** For information about this choice, including consent withdrawal, please see our Privacy Policy

Reviewer #1: **Yes: ** HENK FOLMER

Reviewer #2: **Yes: ** Josh M Colston

---

## [Author Response · Author response to Decision Letter 1]

31 Jul 2025

Response to Reviewers

Reviewer #1: Review of the Manuscript:

“JSTMapp: A web-based joint spatiotemporal modeling and mapping application for epidemiologists”

The manuscript focuses on JSTMapp, a user-friendly web-based dashboard for joint spatiotemporal modelling and mapping of two diseases using Bayesian hierarchical modelling applying INLA and R Shiny. The application deals with cattle bovine tuberculosis (BTB) and human extrapulmonary tuberculosis (EPTB) across Africa from 2005 to 2018.

The study presents a highly a welcome tool for epidemiologists who may not have advanced programming skills. By integrating Bayesian spatiotemporal modelling into a graphical interface, JSTMapp facilitates a complex modelling process.

Strengths

• JSTMapp presents a user-friendly application for complex joint spatiotemporal modelling which is highly relevant and valuable, especially for public health researchers with limited statistical or programming expertise.

• The approach implements multivariate, disease specific and multivariate shared components, allowing for the exploration of spatial, temporal, and spatiotemporal correlations.

• The approach incorporates prediction capabilities for a wide range of prediction intervals.

• The approach includes an interactive exploratory data analysis component, a mapping of standardized incidence ratios module, and a spatiotemporal risk interpolation component enhancing user experience.

• The application is well documented with an easily to use installation guide via GitHub.

Limitations and Areas for Improvement

Despite its strengths, the manuscript and the application have several important limitations that should be addressed:

1. Lack of Model Flexibility

The current version of JSTMapp assumes that all spatially structured, temporally structured, and unstructured components are significant contributors to explaining the spatiotemporal variability of the diseases.

• Critique: In real-world applications, not all random components may be significant.

• Suggestion: The approach should allow model selection options (particularly , selecting spatial effects only, temporal effects only, or spatiotemporal effects only) based on model comparison criteria like the DIC, WAIC, or CPO. See inter alia Jaya and Folmer 2020, J. of Geographical Systems, pp 105-142; 2021, Geographical Analysis, pp 767-817.

Response: We have implemented options for spatial, temporal, spatial + temporal and spatiotemporal. See Figure 2.

2. Fixed Prior Specifications

The application does not allow users to adjust prior distributions for the random effects or the hyperparameters (see Jaya and Folmer 2020, J. of Geographical Systems, pp 105-142).

• Critique: The choice of priors can heavily influence Bayesian model prediction results, especially in the case of sparse data.

• Suggestion: Providing options for users to modify priors (particularly, for spatial precision, temporal precision, and weight parameters) would significantly increase the flexibility and broader applicability of JSTMapp.

Response: We have enabled the choice of precision priors. See Figure 2.

3. Limited Covariate Inclusion

The model is restricted to area and time effects as covariates. No allowance for additional covariates (such as socio-economic factors, climate variables, etc.) is provided.

• Suggestion: Allowing JSTMapp for the inclusion of external covariates would make it more powerful for real-world disease mapping and risk modelling.

• Note: For an overview of the prediction accuracy of multivariate spatiotemporal models with a confounded covariate versus univariate models without covariates, see Jaya and Folmer, 2025, J. of Geographical Systems, 113-146.

Response: We have enabled the inclusion of four covariates as done by similar research by Moraga (2017) [11]. See Figure 2. For illustration see supplementary figure S1 Fig.

4. Lack of Hotspot Detection

The application does not support posterior probability mapping or hotspot analysis (particularly, mapping of areas where the posterior probability of risk exceeding a threshold is above 0.8 or 0.9). See Jaya, Folmer, and Lundberg, 2024, Annals of Regional Science , pp 107-140, for details)

• Suggestion: Adding this feature would be very useful for public health interventions.

Response: We have enabled the hot spot detection (probability maps). See Figures 3 and 4.

5. Future Prediction Needs Accuracy Evaluation

Although the application enables future risk prediction, it does not evaluate the prediction performance (see Jaya and Folmer 2020, J. of Geographical Systems, pp 105-142)

• Critique: Without validating using a training and testing dataset split, the forecasted risks lack an objective measure of accuracy.

• Suggestion: The authors should consider integrating a model validation procedure where part of the data is used for training and another part for testing. Predictive performance metrics such as the MAE, RMSE, or correlation coefficients between observed and predicted values should be provided as statistics to assess forecast accuracy. (See inter alia Jaya and Folmer 2020, J. of Geographical Systems, pp 105-142; Jaya and Folmer, 2025, J. of Geographical Systems, 113-146)

Response: The functionality of forecasting/future prediction has been omitted due to the concern by Reviewer 2, that the future predicted trend seems to fall drastically, which seems unrealistic. We somehow agree with Reviewer 2 concern, actually the sudden fall of future predicted values emanated from the use of CONSTANT average value of the future expected, averaged from the preceding expected values in the observed period, which may not be realistic. We needed to find a way of using non-CONSTANT average, but implementing this at this time is not an easy task. Since future forecasting in many cases is considered optional, we have decided to omit the functionality for now, i.e in the most basic form, joint spatiotemporal models are fitted without forecasting. We could incorporate this in our future updates.

6. Deployment Issues

The application performs better when run locally rather than online via a Shiny server due to memory limitations.

• Suggestion: While the paper recommends using GitHub local installation, applying Dockerization or lightweight server deployment are likely to improve online usability.

Response: We thank the reviewer for this suggestion, but currently we think we can maintain the GitHub useability since other similar publications have used GitHub ( e.g Johnson et al., 2021) [9].

7. Graphics and Interface Improvements

• Critique: Some of the graphical outputs could be improved in terms of better readability (e.g., appropriate scaling legends, improving color schemes for color-blind users).

• Suggestion: Future updates should offer user-friendely color palettes and exportable high-resolution images.

• Note: For high-resolution prediction and mapping the author may consult Jaya and Folmer 2022, J. of Geographical Systems, 525-581.

Response: Our spatial prediction was actually based on the mentioned authors’ articles, including the above-mentioned article. See references [17], [19].

Minor Points

• Typographical and grammatical editing are needed to improve the clarity and professionalism of the manuscript. Some minor errors in English expression should be corrected.

Response: The manuscript was re-edited by an English professional lecturer, Moses Fillimoni, at Nalikule College of Education in Lilongwe, Malawi.

• It would be beneficial if the authors could provide a video tutorial or sample case studies in the GitHub repository for beginners.

Response: We thank the reviewer for this suggestion, but we think can provided in the future. Similar previous research has been published without such supplementary materials (Johnson et al., 2021) [9].

Reviewer #2: The authors describe a bioinformatic application called JSTMapp that they have developed in R, and can be used to fit statistical models to epidemiological surveillance datasets consisting of spatiotemporally referenced case-counts for multiple diseases. JSTMapp can be run using open-source tools such as R, GitHub, or an online interface. They provide an example dataset of national, annual level case-counts of two diseases – BTB, and EPTB – sourced from publicly available repositories and georeferenced to areal units of numerous African countries. When loaded into the JSTMapp interface, the application applies a suite of modeling and data visualization processes to the data allowing the user to explore trends and parameters derived from the model, including spatially interpolated predictions based on shapefiles of the areal units. The example dataset also includes the environmental variables of temperature and precipitation (however it is unclear to me if these are included in the model, as I do not see parameter estimates for them under the “Model estimation” tab).

Response: We have included options for covariates. See Figure 2.

As described, the tool has notable applications for epidemiologists and students of biostatistics, although some upfront work is required of the user to set up the time series data in the right format and link it to the polygons in the shapefile.

I am someone who works with spatiotemporally referenced health data frequently, but who has only intermediate R and basic GitHub proficiency. I have not myself ever fitted the kinds of models employed by JSTMapp (INLA, MCAR etc.) though I am aware of them, have colleagues who work with them frequently, and understand them to be appropriate and state of the art methods for this kind of data. I therefore approach this review from the perspective of the “ordinary epidemiologist with limited analytical skills” that the authors describe.

Introduction:

Not being previously familiar with the term “joint spatiotemporal modeling”, it is unclear to me whether in these kinds of model, each disease outcome serves as a predictor for the others. Is the burden of each disease jointly modeled conditional on the presence of the others. Could the authors please clarify? Is it only applicable to infectious diseases? This is not specified.

Methods:

This section starts with a lengthy explication of the modelling using equations and formal notation. I am not qualified to review these equations, however I do believe that in most journal articles, equations are numbered so that they can be easily referenced within the text.

Response: We have numbered the equations. See pages 2-4.

I think the “components of the application” should be moved up to the beginning of the methods section, and the model descriptions should be lower down or even moved to a supplemental appendix.

Response: In most cases, theory behind the application or software come first and application description follows. We therefore think that we can maintain the status quo. Also, the theory is very important piece of information for the application and we feel that leaving it in the appendix may not be a good idea.

Results:

I’m not sure that a results section is even required for this type of paper, since you are not offering an interpretation of scientific findings, but rather describing a tool and an illustrative example of its application. I will defer to the editor on this point, but perhaps “illustrative example” would be a better name for this section.

Response: We have renamed the results section tittle to “Illustrative example”. See page 6, line 3.

The sources of the illustrative data could perhaps be described in a separate section with its own subheading, and I don’t think that full URLs should be given within the text. Rather, online sources should be cited using in-text citations e.g. (WHO 2025) referencing sources in the bibliography which would include their URLs. Perhaps the sources could be better presented in a small table.

Response: We have included the subheading for data sources. See page 6, line 4.

I was able to download the data and JSTMapp R package and run the application using RStudio. I was not able to do this within GitHub as I am less familiar with that platform. The database that’s downloadable from GitHub contains what appear to be several meteorological variables (temp, tmax, tmin, prec). These are static across years within each country in the database and are not described or cited in the article. It is unclear if they are being used as predictors in the model, and in any case, they probably should not be, since national aggregate statistics are not a suitable level of resolution for weather parameters.

Response: We thank reviewers for their suggestion, but our approach was based on the literature (Arab et al., 2014, http://www.malariajournal.com/content/13/1/126), who also used such weather parameters for national level malaria cases model in West African countries. Nevertheless, we have highlighted the use of country level data as weakness in the discussion section. See page 9, line 13-15. We have also cited the literature to support the use of the country level weather data (See page 6, line 22-24).

Figure 1: I selected the variables in the same way that the authors described, but the visualizations did not appear in exactly the same way. Instead of the maps in the upper left-hand corner of figure 1, I got an error message saying “the following option does not exist: check.and.fix..

Response: We may not be sure of what happened to your end, but it might be due to not uploading or selecting data variables. After data selection such errors could just be ignored and wait for some time for outputs. Sometimes it could be due to uninstalled R packages from the list in Table 1.

Furthermore, the label for figure 1 does not sufficiently describe the map panel. Are these predictions from the INLA component of the model? The legend title is not legible in either the article figure, or in the application interface. I realize that the data is meant to be illustrative, but the authors should perhaps acknowledge that in a real epidemiological analysis, it would not be appropriate to do a spatial interpolation for an entire continent that was based on national level aggregates, since that does not offer sufficient spatial resolution. Similarly, annual totals are probably not a sufficient temporal resolution to derive conclusion about infectious disease transmission. The optimal use case for this package, I think, would be at least monthly or weekly resolution epidemiological data aggregated to province, or district level.

Figure 2: Since the other tabs offer visually appealing visualizations of the model results, could the “Model estimation” tab do the same, rather than displaying raw R model output? Perhaps the parameter estimates could be visualized in dot-and-whisker plots.

Response: We have included the box whiskers plot in the model estimation tab. See Figure 2.

Figure 3: The maps in this figure did not appear when I ran the application in RStudio. I got the same error message about the option not existing.

Response: Basically, all tabs after model estimation tab depend on this tab. So, ensure that the model results are done before clicking on the subsequent tabs. Also, in some cases, it might take at least 3-5mins after clicking on a tab before displaying outputs due to heavy computations. We have included this note in the manuscript and the GitHub repository.

Figure 4: The prediction maps appeared when I ran the application, but the estimation maps did not (same error message).

Response: Also apart from general delayed computations, I suspect one of the required packages listed in the manuscript might not have been installed.

The application does not seem to offer any customizability in the visual outputs and as such is not particularly interactive. More important than generating a series of panels for each year, would be for the user to be able to zoom in and out of the map and click on particular locations to see the prediction values displayed in a pop-out.

Response: We have transformed single maps into the interactive mode. Nevertheless, faceted maps could not be done due to incompatibility of tmap facet functiona

---

## [Decision Letter · Decision Letter 1]

3 Sep 2025

JSTMapp: a web-based joint spatiotemporal modelling and mapping application for epidemiologists

PONE-D-25-13633R1

Dear Dr. Ngwira,

We’re pleased to inform you that your manuscript has been judged scientifically suitable for publication and will be formally accepted for publication once it meets all outstanding technical requirements.

Kind regards,

Rebecca Lee Smith, D.V.M., M.S., Ph.D.

Academic Editor

PLOS ONE

Additional Editor Comments (optional):

Reviewer #1:

Reviewer #2:

Reviewers' comments:

Reviewer's Responses to Questions

**Comments to the Author**

Reviewer #1: (No Response)

Reviewer #2: All comments have been addressed

2. Is the manuscript technically sound, and do the data support the conclusions?

Reviewer #1: Yes

Reviewer #2: (No Response)

3. Has the statistical analysis been performed appropriately and rigorously?

Reviewer #1: Yes

Reviewer #2: Yes

4. Have the authors made all data underlying the findings in their manuscript fully available?

Reviewer #1: Yes

Reviewer #2: Yes

5. Is the manuscript presented in an intelligible fashion and written in standard English?

Reviewer #1: Yes

Reviewer #2: Yes

Reviewer #1: THE AUTHORS HAVE IMPLEMENTED MY COMMENTS AND RECOMMONDATIONS SATISFACTORLY .

Reviewer #2: Thank you for addressing my initial comments from the first round. The limitations are better characterized and the application is more visually appealing now. After a bit of playing around and a few failed attempts (I think due to having the input files stored in different folders) I was able to recreate in RStudio the output shown in the manuscript figures. I will note that the figures are a little slow to load, and that the bottom time series panel under the "Spatial and Temporal Risk" tab did not load, instead giving me the message "plot omitted-your model has no temporal component". Also, the output on the "Correlation" is more limited than in the corresponding figure - perhaps I was meant to select some covariates in the "Model estimation" tab? - and in legends on some of the maps are so large that they obscure the content.

In any case, I think the paper can be accepted, and these minor issues can be addressed while the paper is in press. Congratulations on a well-written paper and an appealing application that has some (admittedly narrow) applications of public health relevance.

**Do you want your identity to be public for this peer review?** For information about this choice, including consent withdrawal, please see our Privacy Policy

Reviewer #1: No

Reviewer #2: **Yes: ** Josh M Colston

---

## [Editor Report · Acceptance letter]

PONE-D-25-13633R1

PLOS ONE

Dear Dr. Ngwira,

I'm pleased to inform you that your manuscript has been deemed suitable for publication in PLOS ONE. Congratulations! Your manuscript is now being handed over to our production team.

Kind regards,

on behalf of

Dr. Rebecca Lee Smith

Academic Editor

PLOS ONE